# Branding Places through Experiential Tourism: A Survey on the Features of the Experiential Product and Enterprises in Greek Regions



**Athena Yiannakou** [1,*] **, Angelina Apostolou** [1] **, Vasiliki Birou-Athanasiou** [1] **, Apostolos Papagiannakis** [1] **and Athina Vitopoulou** [2]

1   School of Spatial Planning and Development, Aristotle University of Thessaloniki, 54124 Thessaloniki, Greece; aaaposto@plandevel.auth.gr (A.A.); vmpyroua@plandevel.auth.gr (V.B.-A.); apa@plandevel.auth.gr (A.P.)

2   School of Architecture, Aristotle University of Thessaloniki, 54124 Thessaloniki, Greece; avitopoulou@arch.auth.gr

*   Correspondence: adgianna@plandevel.auth.gr

**Abstract:** The focus of experiential tourism is for the consumer or visitor to experience the tourist destination and to actively interact with local people, cultures, and the place itself. In fact, it can be seen as a form of tourism that builds upon place identities, both tangible and intangible, by energetically introducing the visitor to the culture, history, nature, traditions, cuisine, and social life of a place. In doing so, the emotional, physical, or spiritual experience of the consumer becomes a dynamic source of place branding. The paper investigates the main features of experiential tourism in the Greek regions of Central Macedonia, and Eastern Macedonia and Thrace, and discusses their interactions with place identity. Our methodology consists of a qualitative survey based on semi-structured interviews with stakeholders and a thematic analysis to trace the main features of the experiential product and enterprises that develop such products. The paper concludes that experiential tourism in Greece bears many of the features highlighted in the literature. Furthermore, our findings underline some new aspects, especially the links between the experiential product, small and well-qualified enterprises, and a place's tangible and intangible identities, which make experiential tourism an opportunity for locales and their branding.

**Keywords:** experiential tourism; Greek Black Sea regions; tourism product; tourism enterprises; place identities and branding

## 1. Introduction

Experiential tourism has become increasingly attractive to travellers during recent years, while the post-COVID-19 period is expected to bring it to the forefront of tourism practices as an innovative and sustainable form of tourism, capable of supporting local societies in their quest for sustained development.

According to the well-known study of Pine II and Gilmore [1] on experience economy, an experience is not an amorphous construct; it is as real an offering as any other service, good, or commodity. Although, in many respects, everything entails an experiential dimension, in the field of economy, experience is designed, intentionally produced (staged), and eventually priced [1,2]. Undoubtedly, one of the pioneer examples of the experience economy is tourism [2,3], an industry where the distinction of experience as a separate, valuable commodity offers considerable economic opportunities. The tourism industry provides experiences that are not only different from everyday life, but have strong emotional resonance, create deep involvement and strong connections, and also have the potential to transform consumers [4].

"Experiential tourism", as defined by several scholars, or "creative tourism", as defined by others, began to appear in the second half of the 1990's as a new tourism strategy

paradigm that contradicts mass tourism [2,5,6]. It is a strategy that is built upon authentic experience and connects the traveller with the essence of a place through active participation and interaction with the local people, cultures, and environment.

Tourism experiences are influenced by various factors [7]. For instance, Pine II and Gilmore identified four realms of experience: entertainment (passive, absorption), educational (active, absorption), escapist (active, immersion), and aesthetic (passive, immersion) [1]. Researchers have explored other dimensions that affect tourism experience. Factors such as environment aesthetics, emotions, active participation, price, tourist satisfaction, and memorability have been strongly connected to tourism experience [8]. Furthermore, experience emerges from the interaction between destination and traveller/tourist—with destination functioning as the "theatre" at which experience takes place, and tourist as the "actor" who has to play his/her own role [2]. A positive experience is strongly related with higher tourist satisfaction and a favourable and enhanced destination image [9].

Destination image is one of the most important criteria for selecting a place to visit, and is also a challenge for destination management organizations [10]. Destination image has been defined as the sum of beliefs, ideas, and impressions that a person has of a destination [11]. As such, a destination is more than a place. It is a complex of spatial units and different semantic layers, and a combination of all the products, services, and ultimately, experiences provided locally [10,12]. Regions and cities have been globally characterized by a plurality of efforts to define their vision, construct their identity, and shape their images to become more attractive and competitive and to develop efficient tourism strategies [13]. An effective and strategic way to construct a positive image of a place, based on its distinctive characteristics, and to become competitive among other places, is through place or destination marketing and branding [14,15].

The implementation of place marketing largely depends on the construction, communication, and management of a place's image since encounters between places and their users take place through perceptions and images [16]. On the other hand, as argued by Deffner et al. [16], place marketing cannot be anything other than "the conscious and planned practice of signification and representation", which constitutes the starting point for examining place branding.

Destination branding is a specialized form of tourism-oriented strategic marketing communication [17], an intangible resource that contributes to the growth of a place, a unique asset for tourism, and a source of competitive advantage [15]. Successful destination branding needs to be anchored by the identity and authenticity (staged or real) of the destination [15]. In other words, the starting point for developing a destination brand is the place's identity [18], which consists of (a) tangible elements including food, accommodation, heritage sites, transport, town planning, and architecture; and (b) intangible elements, which include hospitality, visitors' information, culture, practices, history, religion, people, politics, environment, entertainment, and security [15]. Therefore, experiential tourism can be seen as a strategy that builds upon place identity, both tangible and intangible, by actively introducing the consumer to the culture, history, nature, traditions, cuisine, and social life of a place. In doing so, the emotional, physical, or spiritual experience of the consumer has become a dynamic source of place branding. From this point of view, on-site value creation processes become the core foundations that the tourism industry must acknowledge to plan, develop, involve, and accommodate tourists, so that they are able to actively partake in experiential tourism practices [19].

This paper presents the results of research that explores the features of experiential tourism in two Greek regions which fall within the wider cross-border Black Basin region. The research is part of the EU (Black Sea Cross-Border Cooperation) Project PRO EXTOUR: Promoting Heritage- and Culture-based Experiential Tourism in the Black Sea Basin. In Section 2, based on the relevant literature, we elaborate on a conceptual framework regarding the main features of experiential tourism and formulate our main research questions. Section 3 provides a brief profile of the study area, as well as of the role of the tourism sector, and describes the main methodological steps. Section 4 presents the main

findings of the research and, finally, in Section 5, we provide a discussion and portray our main conclusions, with a special focus on how the main features of experiential tourism, as traced in our study area, interact with place identities.

## 2. A Conceptual Framework of the Main Features of Experiential Tourism

Experience has always existed in destinations, but it was considered a context rather than a content [2]. A destination, as a complex phenomenon that represents a large entity consisting of a variety of tangible and intangible components [15], may serve as the locus for multiple experience settings [2]. Thus, a tourism destination can be interpreted as an amalgam of goods, services, and production units which offers a tourism experience to individuals or groups of people, in order to satisfy their tourism needs or desires [20]. Experiential tourism includes the story of a destination and encompasses several features related to experiential products and to the involved enterprises. The academic literature has explored a variety of factors that are affecting tourism experience and that are interwoven with experiential tourism from various perspectives. From this literature, we may identify five main features of experiential tourism.

The first feature is that experiential tourism, being complex by nature [3], comprises a variety of dimensions. This variety concerns the tourism product, services, and enterprises, as well as tourism and traveller categories—including cultural, eco, educational, heritage, nature tourism, etc.—where activities are environmentally sensitive, displaying respect for the culture of the host area, and looking to experience and learn, rather than to merely stand back and gaze [5].

The second feature relates to memory and senses. Whereas goods are tangible and services intangible, experiences are memorable [5]. Experiential marketing is interested in those spheres of consumption where consumers seek to experience services that engage their emotions and, therefore, bare strong symbolical and aesthetical values [4]. Thus, tourists no longer seek simply to purchase goods and services, but to receive memorable, sense-arousing, satisfying experience encounters [21]. Experiential tourism entails a complete internal emotional immersion of an individual into an object of aesthetic appreciation; at the same time, aesthetics are an experiential process that stimulate all five senses [8]. Sensorial and emotional experiences tend to be remembered better by individuals and are more memorable if they induce feelings of pleasure and arousal [21]. These emotional impressions of experience, by consuming a tourism product or encountering a tourism service, are strongly associated with vivid memories and feelings of place attachment and revisiting intentions [21].

The third feature pertains to the active participation of the traveller. In the age of experience economies, active participation seems to have been highly connected to tourism experience [8]. According to Pine and Gilmore, tourism experience occurs when a service is performed uniquely and memorably, involving the customer as an active participant [1]. The customer seeks to be involved in new experiences that may actually affect his/her life, rather than simply packing his/her schedule with a variety of entertaining experiences [6]. The whole process of active participation is innovative, compared to the conventional tourist product, as it is a creative and knowledge-creating process, based on the interaction, involvement, and immersion between place, theme, and tourist [2,5,8] that leads to the self-development of every tourist who participates in experiential activities [6].

The fourth feature, common to the experience economy in general, is that experiential tourism is strongly related to the notion of customization [5,8]. In an experience-based activity, the tourist enters into a multifaceted interaction with the actors and the setting of the experience [2]. This means that the experience includes, among other aspects, the people one meets, the places one visits, the accommodations where one stays, the activities one participates in, and the memories one creates [5]. It encourages visitors to actively participate in the experience and promotes activities that draw people outdoors, and into cultures and communities [5]. So, the creation of an experience activity presupposes the

accumulation of detailed information about tourist tastes, preferences, and values [2], as experience is very personal and individual [5].

Finally, a fifth feature is the dimension of staged authenticity. Despite the fact that tourism cannot be universally authentic, experiential businesses tend to package the offered experiential product or service in such a way that customers perceive it to be authentic [1,21]. In this context, it has been found that well-staged experiences lead to enriched and vivid visitor memories [21]. In order to achieve this staged authenticity, experience should be designed, intentionally produced (staged), organized, and foreseen by tourism enterprises [2]. The staged authenticity of experience-based practices involves a strategically designed scenario, related to the creation of myths or narratives that have to be constructed through a maze of continuity and discontinuity, built on the capabilities and competences of the destination [2]. It should again be noted here that, despite the fact that tourism cannot be universally authentic, experiential enterprises tend to package the offered experiential product or service in such a way that customers perceive it to be authentic [1,21].

Following the above conceptual framework, our main research questions are: What are the specific features of the experiential tourism product in a study area with a pronounced mass tourism profile? What are the specific features of the involved enterprises, and what are the main problems that experiential tourism is faced with in the study area? Finally, how does experiential tourism interacting with place identities thus contribute to its branding?

## 3. Case Study Area and Methodology

### 3.1. A Profile of the Study Area

The Greek part of the cross-border Black Basin region consists of two administrative regions, the regions of Central Macedonia, and Eastern Macedonia and Thrace (from now on, BSB-GR). The BSB-GR regions cover an entire area of 33,352 km$^2$ in the north of the country (Figure 1), with a total population of 2.5 million people, accounting for 23% of the national population. The Region of Central Macedonia is the second-largest region in Greece, after Attica, with a population of 1,882,108 people, accounting for 17% of the entire Greek population, 58.7% of which live in the Regional Unit of Thessaloniki. To a large extent, it is an urban region, with the city of Thessaloniki, the second largest metropolitan area in the country, being its main urban centre. The region of Eastern Macedonia and Thrace is one of the smallest regions in Greece, with a population of 608,182 people, or 5.6% of the country's total population.

Following the tourism patterns and trends of the entire country, both BSB-GR regions have a well-developed touristic profile, especially regarding their coastal areas, comprised predominately of mass summer vacation tourism and, to a lesser extent, of other, "alternative", non-mass forms of tourism. In Greece, tourism, through its various sectors and branches, has a very crucial contribution to the national economy. In 2019, i.e., before the COVID-19 pandemic, the direct contribution of tourism as a percentage to the national GDP was 12.5%, while the total contribution of tourism reached 33% of the national GDP [22]. As expected, amongst Greek regions, two island ones, the South Aegean and Crete, concentrate the highest share of incoming tourism revenues, almost half of the total incoming revenues. The share of the BSB-GR regions in the incoming tourism revenues is 15%, a figure that represents 13% of the contribution to their total GDP. The region of Central Macedonia records the fourth-highest share in the incoming tourism revenues in Greece (13%), and the region of Eastern Macedonia contributes 2% [22,23].

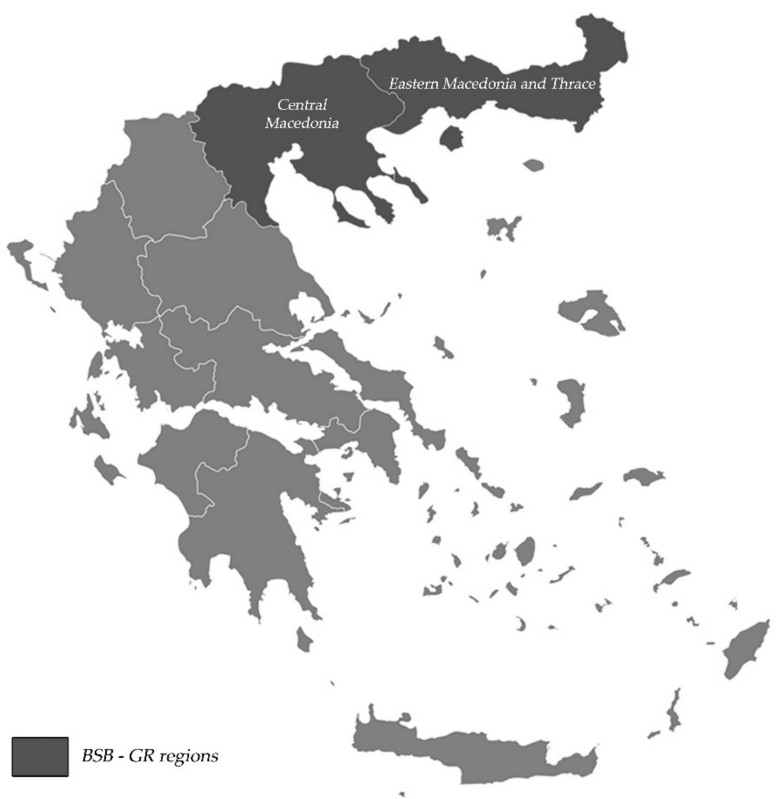

**Figure 1.** Study area.

According to national statistics [24], in 2019, regarding arrivals, the region of Central Macedonia contributed to Greece's tourism scene by a share of 12%. The Regional Units of Thessaloniki and Halkidiki are the two major tourist destinations in the region. Throughout 2014–2019, the region marked a considerable increase in arrivals, of 45%, though this is lower than the national average. The region of Eastern Macedonia and Thrace plays a rather limited role in the country's tourism scene, as it concentrates just 3% of the county's total arrivals in tourist accommodations. In 2019, more than half of the arrivals were recorded in the Regional Units of Kavala–Thasos (51.2%). However, the region achieved a significant increase of more than 30% in arrivals during 2014–2019. The region of Central Macedonia concentrates more than 10% of the total overnight stays in the county, while Eastern Macedonia and Thrace concentrates just 2.3%. During 2012–18, overnight stays increased in both regions.

*3.2. Methodology of Survey Design and Thematic Analysis*

The research was based on a qualitative survey consisting of in-depth interviews and panel discussions that enabled us to deepen our understanding of the special features of experiential tourism, both in terms of the experiential product and the enterprises involved, in a study area dominated by mass tourism forms. The survey was conducted between October 2020 and January 2021. Taking into consideration the spatial scale of the study region and the sector in question, all partners of the EX-TOUR project decided that approximately 25 interviews in each study area would be adequate for the research. The selection of the interviewees was a crucial issue, so as to focus on stakeholders that were either involved in experiential tourism activities and tasks, or that were all-important in promoting such tasks (Figure 2).

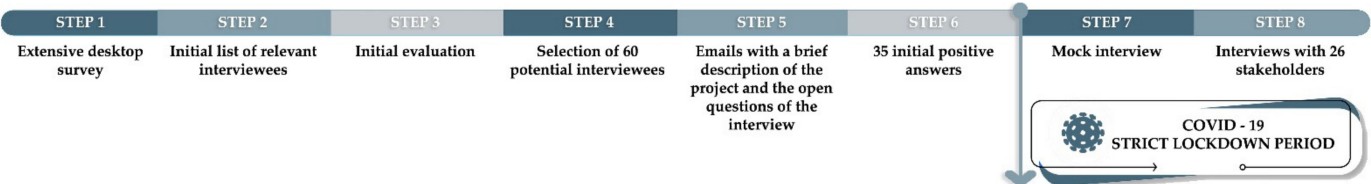

**Figure 2.** Survey design.

At an initial stage, we performed an extensive desktop survey, aiming to identify relevant stakeholders as follows: (a) stakeholders both of the private or the public sector who were involved in the wider field of alternative forms of tourism and in various subsectors (accommodation, restaurants, tour agencies, government, public organizations, destination management organizations, educational institutions); (b) stakeholders, mainly of the public sector, who are involved in policy making at the local, regional, and national levels. An initial list of relevant interviewees was completed (Table 1), representing all the categories of stakeholders that were identified by the PRO EXTOUR project. This list, which was practically open, eventually included 124 stakeholders. After an initial evaluation with the help of our theoretical framework, we selected 60 potential interviewees who seemed to be more closely involved with experiential activities in the field of culture and heritage. This initial evaluation concerned primarily tourism enterprises, from both the private and public sectors. For these enterprises, we checked whether the description they provided for their products/services seemed to meet at least some of the features described in our conceptual framework. For a better preparation of the informants, and in order to select the ones who were oriented towards experiential tourism tasks the most, i.e., their products/services met at least some of the important features described in the conceptual framework, emails were sent including a brief description of the project and the open questions that would be used during the interview. It should be noted here that certain stakeholders of the public or corporate sector, such as the Ministry of Tourism, the Greek National Tourism Organization, the Greek Tourism Confederation, the Hellenic Hoteliers Federation, and the two regional administrations were chosen in an ad hoc manner, as they play a crucial role in the tourism sector at both the national and regional level. On the whole, we received 35 initial positive answers. It is worth noting that the entire survey period coincided with the most severe lockdown of the COVID-19 pandemic period held in Greece. Therefore, this was a period of uncertainty for many of the private companies that were contacted, as they either were shut down or very reluctant to take part in the survey. This situation led to an extension of the interview phase. Finally, 26 stakeholders participated in the in-depth interviews, according to their relevance and their availability to participate.

**Table 1.** Number of potentially relevant stakeholders per category.

| Category | Number |
|---|---|
| Public Organizations (National, Regional, Local level) | 40 |
| Private Companies (Accommodation, Experience Activities/Ventures, Tourist Offices, Thematic Tours, Restaurants/Gastronomy) | 50 |
| Other | 34 |
| Total | 124 |

Before launching the 26 interviews, we decided to perform a mock interview with an entrepreneur well-versed in experiential tourism from a region outside our study area, whose company provides a high-quality experiential product. During this interview, several factors were tested, including the validity and the understanding of the main questions.

This mock interview provided useful feedback regarding an overall state of the art of experiential tourism in Greece, and contributed to improvements to the main questions, the communication skills with the other interviewees, and the overall quality of the survey.

The final sample of 26 stakeholders included 15 private and 11 public sector stakeholders (enterprises and organisations) that are either active in the field of experiential tourism or influence policy making in the field, such as regional administration, local authorities, and sectoral organisations (Table 2). The interviews were conducted using an interview guide, which was the same in all countries participating in the PRO EXTOUR project, and was translated into Greek to further assist the process. The questions were adjusted, when necessary, based on the category of the stakeholder being interviewed. In addition, we used open questions in order to identify how the stakeholders understood experiential tourism, without directing their opinion with any predetermined possible answers. The interviews lasted between half an hour to two hours, depending on the informant and the pace of the interview. Due to the restrictive measures of the lockdown period, all the interviews took place remotely. Throughout the course of the survey, the members of the research team continued to have weekly and additional ad hoc meetings to debrief each other regarding the achievements and progress of the study, to discuss the challenges faced, and to share any interesting information that might have emerged from the interviews.

**Table 2.** Number of interviewed stakeholders per category.

| Category | Number |
|---|---|
| **Public sector** | 11 |
| National and Regional government bodies | 4 |
| Destination management organizations | 2 |
| Other public organizations | 5 |
| **Private sector** | 15 |
| Private enterprises: Experience Activities/Ventures, Accommodations, Thematic Tours, Gastronomy | 15 |
| Total | 26 |

Finally, based on the conceptual framework presented in Section 2, which was used as hermeneutical tool, we conducted a deductive, thematic analysis of the interviews with the objective to track, recognize, and evaluate (a) the extent to which the main features of experiential tourism are evident in our study area, and (b) the main features of the involved enterprises and their marketing approaches. Adapting a qualitative graded scale already used and tested in another study [25], we ranked the main features of the experiential tourism product in our thematic analysis according to the frequency and the emphasis of references which were made to them. This graded scale is as follows:

- no reference: when the feature was not mentioned at all;
- weak reference: when only a few stakeholders referred to a feature, either explicitly or implicitly;
- weak reference, but from the more experiential-oriented enterprises: when a feature was mentioned explicitly by few stakeholders, among which were enterprises providing experiential product closer to the main features of experiential tourism;
- strong reference: when explicit references to a feature were made by many stakeholders.

In addition to the thematic analysis conducted for the total of 26 interviews, where deemed necessary, we focused on 16 interviews concerning the enterprises, both of the private and public sector, which were ranked as the most oriented to the experiential product.

## 4. Results

### 4.1. Main Features of the Experiencial Tourism Product

Figure 3 summarizes the main findings for the features of the experiential tourism product, as drawn by the thematic analysis of the 26 in-depth interviews with stakeholders of the private and public sectors.

| | | | FEATURES | | | | |
|---|---|---|---|---|---|---|---|
| | | | Complex & Integration of multidimensional parameters | Senses, emotions, memory | Active participation | Customization | Staged authenticity |
| CATEGORY/ SECTOR | PUBLIC | Policy making bodies | ● | ○ | ○ | ○ | ○ |
| | | Destination management organizations | ● | ○ | ○ | ○ | ○ |
| | | Museums | ● | ● | ● | ● | ○ |
| | | Ephorates of Antiquities | ● | ● | ● | ● | ○ |
| | PRIVATE | Experience Activities/Ventures | ● | ● | ● | ● | ● |
| | | Accommodation | ● | ● | ● | ● | ● |
| | | Thematic Tours | ● | ● | ● | ● | ● |
| | | Gastronomy | ● | ● | ● | ○ | ○ |

*Number and type of reference*

○ No reference

◔ Weak reference

◑ Weak reference, but from the more experience-oriented enterprises

● Strong reference

**Figure 3.** Features of the experiential tourism product, as drawn from the 26 interviews with stakeholders.

The findings of the qualitative survey show that the experiential tourist product in the BSB-GR regions is a dynamic combination of various elements. There are several dimensions that diversify the experiential tourism product from the traditional one, among which the most crucial ones fall, with less or more emphasis, on the five features as recorded in the conceptual framework: the integration of multi-dimensional parameters, the stimulation of the senses, emotions, and memory, and active participation are the most apparent features, whereas customization and stage authenticity are also present, although with less frequency than the first three features. It is worth mentioning that these last two features are mostly present at the more experience-oriented enterprises of the private sector.

In both BSB-GR regions, experiential tourism is confronted by the stakeholders as a "general term" that integrates activities and services in the fields of culture, gastronomy, rural, and adventurous experiences.

In the cultural field, the main products/services concern alternative guided tours, themed walks, and routes in the city and nearby archaeological sites. In some cases, these products are combined also with physical activities such as biking, wine and food tasting, and local cooking lessons. It is worth mentioning that sites selected for visiting are not necessarily historical landmarks or well-known historical places of the city, but spots with a hidden history. In such experiential activities, the enrichment of the tourism product is achieved by provoking aspects such as fantasy and play. Visitors are invited to uncover

mysteries or solve enigmas, and, in several cases, historical information is provided through small, minute-long performances by actors.

In the context of gastronomic tourism, the main products concern wine trails and wineries that introduce visitors to local culture and tradition through wine-tasting, but also with their active participation in the wine production process, depending on the season (for example vine–harvest). Sometimes, cooking workshops are organized for the visitors by using not only ingredients from the vineyard, but also other local ones.

In the field of agricultural tourism, in both BSB-GR regions, activities related to tourism experiences have been developed mostly in rural, remote areas. This is justified by the fact that, on the one hand, these areas need to develop complementarily to the primary sector activities to become more competitive within their wider territory. On the other hand, rural territories often present a rich set of unique natural resources that allow "authentic" and memorable tourist experiences. Experiential activities related to this field are essential oil distilleries, visits to farms and olive groves, participation in the agricultural and beekeeping activities, and tours about the agricultural way of life. It is worth mentioning here that most of these experiences cannot be categorized in the basic, existing tourism sectors such as accommodations, restaurants, tour agents, tour guides, etc. In the rural areas, many physical and adventurous activities, such as trekking, horse riding, river trips with canoes–kayaks, rafting, and mountain biking, are ventures whose primary scope is to stimulate experiences based on physical activities. Some of these activities are also connected to other local cultural activities, such as mushroom and traditional herb hunting.

An important number of the experiential activities that were recorded in the qualitative survey provide the stimulation of senses and the generation of emotions, as well as physical activity. In other words, to a large extent, the tourism experience product that was recorded is well oriented to stimulate the senses, the heart, the mind, and the body. Another key element that also emerged in both BSB-GR regions is memory. The interviewed tourism stakeholders seem to adopt a perspective that linking memory with local culture, through cuisine, for example, enhances the visitors' experience and influences their ability to recollect past travel experiences or to retrieve vivid information about them.

To achieve a memorable experience, the outstanding role of active participation is underlined in contrast to passive entertainment. To accomplish that, the fewer the participants, the more authentic the experience is perceived as being. Therefore, most of the activities take place in small groups. Here, it is worth mentioning that, although active participation is integrated in the offered experiential products and activities, co-creation processes between enterprises and visitors are absent.

The customization feature appears to be rather weak and undervalued by the majority of the interviewed stakeholders. However, it is worth noting that the more experience-oriented enterprises of the private sector in the BSB-GR region are strongly associated with customization. This means that their tourism product is designed by focusing on the visitor's specific preferences, interests, values, and needs. So, the more the offered product is experience-driven, the more customer- and local community-oriented its development is, focusing on the visitor's values and needs.

Tourism enterprises related to experiential tours, walks, and activities pointed out that their activities have to appeal to the tourist's fond desires and imaginative associations. To do so, they must draw on myths, histories, and fantasies that are mainly associated with the local culture and history of the region. However, considering that visitors are on a "quest for authenticity" on their trips and experiences, the most successful tourism enterprises, especially of the private sector, offer "staged authenticity" to promote locality through storytelling techniques. In addition, to offer a high-quality experiential product, they highlighted the need for a strategically designed scenario, with emphasis given to the surroundings and the triggering of senses, feelings, emotions, and thoughts.

### 4.2. Features of Experiential Tourism Enterprises

The overall desktop research that took place as a first step of our qualitative survey showed that, despite the growing attention to experiential tourism internationally, only a small number of enterprises related to tourism have actually developed experiential activities in both BSB-GR regions. Figure 4 summarizes the main features and marketing approaches of the tourism enterprises per category and sector of activity, as drawn from the 16 interviews with stakeholders from experiential tourism enterprises. The categories we used for both the enterprise features and marketing approaches resulted from the thematic analysis of the in-depth interviews. It should be noted that one of the aims of the qualitative survey was to investigate, with general questions, the features of experiential tourism enterprises, especially their size, year of establishment, special skills of the workforce, and the marketing approaches they follow. The similarities between the answers of the interviewees, and their correlation to the conceptual framework, resulted in the categories used below, in Figure 4.

| | | | FEATURES | | | | MARKETING APPROACHES | | |
| --- | --- | --- | --- | --- | --- | --- | --- | --- | --- |
| | | | Newly established businesses | Small workforce | Skills | | Word of mouth | Social Media | Travel fairs, exhibitions, conferences |
| | | | | | Storytelling techniques | Strategically designed scenario | | | |
| CATEGORY/ SECTOR | PUBLIC | Museums | No reference | No reference | Weak reference, but from the more experience-oriented enterprises | Weak reference, but from the more experience-oriented enterprises | Weak reference | Weak reference, but from the more experience-oriented enterprises | Weak reference, but from the more experience-oriented enterprises |
| | | Ephorates of Antiquities | No reference | No reference | Weak reference, but from the more experience-oriented enterprises | Weak reference | Weak reference | Weak reference, but from the more experience-oriented enterprises | Weak reference, but from the more experience-oriented enterprises |
| | PRIVATE | Experience Activities/Ventures | Weak reference, but from the more experience-oriented enterprises | Strong reference | Weak reference, but from the more experience-oriented enterprises | Weak reference, but from the more experience-oriented enterprises | Strong reference | Weak reference, but from the more experience-oriented enterprises | Weak reference, but from the more experience-oriented enterprises |
| | | Accommodation | Weak reference | Strong reference | Weak reference, but from the more experience-oriented enterprises | Weak reference, but from the more experience-oriented enterprises | Strong reference | Weak reference | No reference |
| | | Thematic Tours | Weak reference, but from the more experience-oriented enterprises | Weak reference, but from the more experience-oriented enterprises | Weak reference, but from the more experience-oriented enterprises | Weak reference, but from the more experience-oriented enterprises | Weak reference, but from the more experience-oriented enterprises | Weak reference, but from the more experience-oriented enterprises | Weak reference, but from the more experience-oriented enterprises |
| | | Gastronomy | No reference | Weak reference | Weak reference, but from the more experience-oriented enterprises | Weak reference, but from the more experience-oriented enterprises | Strong reference | Weak reference, but from the more experience-oriented enterprises | No reference |

*Number and type of reference*

◯ No reference

◓ Weak reference

🔵 Weak reference, but from the more experience-oriented enterprises

⬤ Strong reference

**Figure 4.** Main features and marketing approaches of experiential enterprises.

Among the interviewed enterprises, only a few have a deep understanding of the definition and dimensions of experiential tourism. These are newly established businesses of the private sector which are related to alternative tours and experiences and that focus their activities exclusively on the experiential sector. In other words, their philosophy is to create unforgettable memories for visitors, related to the local history and way of life. All the other interviewed tourism businesses are existing accommodations or tourism enterprises that have shifted from offering mass tourism services to services which incorporate physical activities such as horse riding, hiking, and walks in nature. It should be noted here that there is a gap between the existing and conventional categorisations of tourism enterprises, as some of the existing businesses in the experiential sector could not be classified in the existing tourism categories (accommodation, restaurants, tour agents, tour guides, public institutions, etc.).

As expected, the more experience-oriented an activity or a product provided by an enterprise is, the deeper its understanding of the term "experiential tourism" will be. Indeed, as observed during the empirical survey and the in-depth interviews, a good understand-

ing of the term by an enterprise reflects and results in better products and services. This conclusion was drawn up from the analysis of a few open questions which investigated how stakeholders perceived the term "experiential tourism". The good understanding of the term was evaluated based on the similarities between a stakeholder's perception of the experiential product/service with that of our conceptual framework. On the contrary, some enterprises, particularly those involved in tour guides and accommodation, seemed to miscomprehend the term, and therefore often offer products where the experiential aspect is less prominent.

The most experiential-oriented enterprises with a fully designed tourism product are related to alternative tours and agricultural and wine activities, and are located mainly in the region of Central Macedonia. In the rural and remote areas of the BSB-GR region, appealing and distinctive experiential offerings have been developed by accommodation enterprises to survive in a competitive context. It should be noted that gastronomy experiential activities, such as food and wine tasting and traditional cooking lessons, are organized by wineries, agricultural businesses, and traditional cooperative organizations. This reflects a lack of involvement in this sector of the restaurant industry in both BSB-GR regions.

The survey revealed a huge gap regarding the philosophy and pricing policy between experiential tourism enterprises and travel agencies. Well-organized experiential enterprises acknowledge that a high-quality product (fully designed with a qualified personnel) leads to higher pricing than mass tourism services. At the same time, travel agencies in Greece, in contrast to international travel agencies, regard the offered experiential product as overpriced. This proves a lack of cooperation between these sectors and a low level of knowledge of the experiential product's high-quality importance.

Regarding the basic skills of the employees in the experiential tourism sector, the interviewed stakeholders pointed out that a successful experiential tourism product requires deep and specific knowledge of the activity, as well the subsequent skills needed for it, the history and culture of the place, and the dimensions of experiential tourism. An interdisciplinary approach also seems to be a key element to design, package, and deliver the resulting experiential content. Constant passion, inspiration, and creativity about the activity's subject are basic features for an employee in an experiential tourism business. Other crucial requirements also include very good knowledge of foreign languages, as well as social and communication skills (such as storytelling) for shaping a friendly and interesting cultural contact, and for promoting Greek hospitality, which comprises a major asset of Greek culture. Finally, a sustainable tourism product should adapt to the global competitive environment. Therefore, employees should constantly be trained and educated in new tourism trends and challenges. All these factors are crucial for the production of a high-quality experiential tourism product.

The concept of an experiential product has been implemented mainly by newly established businesses with a small workforce that comprises mostly family businesses. The promotion techniques of experiential products focus on digital marketing through social media, websites, and web platforms. However, according to the surveyed enterprises, the "word of mouth" communication is the most important and efficient marketing practice. Apparently, it was also mentioned that internet allows the traveller to share her/his experience during the whole process. This socialization of the travelling experience entails an opportunity to promote the company when it has met or exceeded the expectations of its clients, but at the same time, there is the risk of defamation, in the case of an experiential product that does not fulfil visitors' expectations.

### 4.3. Main Problems and Needs

Figure 5 summarizes the main problems of experiential tourism in the study area, as drawn from the 26 interviews with stakeholders. The categories used to classify the main problems resulted from the thematic analysis of the in-depth interviews, and more specifically, the frequency of similar answers regarding the problems they are confronted with as experiential tourism enterprises or as those who identify as involved stakeholders.

| | | | PROBLEMS | | | | |
|---|---|---|---|---|---|---|---|
| | | | Functional problems in cultural infrastructure | Lack of cooperation in national, regional & local level | Lack of long-term strategic frame-work & planning | Insufficient transportation system | Lack of clear content definition |
| CATEGORY/ SECTOR | PUBLIC | Policy making bodies | ○ No ref | Weak | Weak | ○ No ref | Weak |
| | | Destination management organizations | Weak | Weak | Weak | ○ No ref | Weak |
| | | Museums | ○ No ref | Weak exp | Weak | Weak | ○ No ref |
| | | Ephorates of Antiquities | ○ No ref | Weak exp | Weak | Weak | ○ No ref |
| | PRIVATE | Experience Activities/Ventures | Strong | Strong | Strong | Weak exp | Weak exp |
| | | Accommodation | Weak exp | Weak | Strong | Weak exp | Weak exp |
| | | Thematic Tours | Strong | Weak exp | Weak exp | Strong | Strong |
| | | Gastronomy | Weak exp | Weak exp | ○ No ref | Weak | Weak |

**Number and type of reference**

○ No reference

◦ Weak reference

◦ Weak reference, but from the more experience-oriented enterprises

● Strong reference

**Figure 5.** Main problems of experiential tourism in the study area.

Among the main problems raised by the interviewed stakeholders are the multiple functional problems of many cultural sites, such as poor maintenance, lack of cooperation within and between all governance scales (national, regional, and local), and insufficient strategic planning, along with a lack of content definition. Such problems often result in deficits in obvious prerequisites of the tourism product: the stakeholders pointed out, for example, that many of the main archaeological and historical monuments, and especially those in distant areas, are often closed on days with higher numbers of travellers, such as weekends.

Of special importance are also the problems related to the transportation system, a significant precondition for travelling. There is lack of integration of different transport modes and poor connection between distant geographical areas with the central ones by public transport. It is worth noting here that, regarding accessibility, in some of the existing activities, especially related to alternative tours, particular attention is given to tours specifically designed for the disabled, the elderly, and people with fewer opportunities by well-qualified specialists. All these actions are undertaken by private tourist companies.

From a marketing and management perspective, aspects of interaction and network creation are important. Until today, only the well-organized businesses have developed strategic partnership networks with large companies, local enterprises, and institutions of the region. However, in general, research has shown a lack of cooperation in the field of experiential tourism at all levels (national, regional, and local). A high-quality experiential product and service, through information sharing and synergies, requires the efficient cooperation of all stakeholders involved in tourism. In particular, collaboration between the public and private sectors would maximize the touristic benefit.

Cooperation at all levels, such as between the private and public sectors, or between the main levels of government, is recorded as one of the major needs. A different philosophy between stakeholders regarding the content and importance of experiential tourism, the lack of local, trained staff, and the lack of financing were pointed out as the main problems that experiential enterprises are faced with. Constant feedback and adaptation to unforeseen or urgent conditions, strengthening existing infrastructure, and the modernization of legislation were also recorded as critical needs.

## 5. Discussion and Conclusions

Greece belongs to those countries that have long relied heavily on a "sun–sea–antiquities" tourism product model, which is characterized by seasonal and geographical concentrations of tourists, mainly during summer, at seaside resorts and, especially, island destinations, which receive the majority of incoming tourist arrivals [26,27]. Thus, the Greek tourism product, composed of infrastructure and services, is still primarily addressed to the mass tourism model [28]. Multiple coastal and insular locations have become well-known destinations for leisure tourism and recreation. In these locations, the tourism and recreation sectors represent a diverse group of enterprises, ranging from travel agencies, hotel businesses, restaurants, and car rental companies, to retail shops, agricultural goods suppliers, and other goods and services providers [29].

Despite the particular importance of the "sun–sea–antiquities" tourism product in the national and regional economy, the findings of our research in the two BSB-GR regions, namely, Central Macedonia and Eastern Macedonia and Thrace, show that selective forms of tourism are also gaining importance in the tourism market. It seems that the new competitive environment, and the dynamic character of tourism, contributes to a shift from the conventional understanding of a destination as a "sun–sea–antiquities" pattern, to the concept of destination as an overall and integrated experience.

Critical findings of our qualitative survey reveal that experiential tourism is still a new field in Greece; yet, these findings highlight the fact that part of the existing experiential product has characteristics, which are designed, intentionally produced, and built upon authentic experience, connecting a traveller with the essence of a place as mentioned in the relevant literature [1,2,6,21]. Overall, the basis of experiential tourism in the study area is its anthropocentric approach, as its product focuses on the active and true connection with the local community, while at the same time, it utilizes each place's tangible and intangible qualities [2,5,8]. The personalized encounters between local people and visitors, as provided in some themed walks, alternative routes, and experiences, plays a critical role in the quality of the tourist product and experience. Furthermore, according to the most specialized of the experiential tourism stakeholders, the hospitality and friendliness of Greek people are key elements in the creation of unique, "authentic", and memorable tourist experiences. All these characteristics of the experiential product have also been pointed out by other relevant research in Greece [5,8,16].

The fact that most of the experiences designed or offered do not fall under the mainstream tourism product categories is an interesting finding that deserves special attention regarding tourism policies. Experiential activities documented in rural areas concern products that focus on experiences from adventures, environmental concerns, and new aesthetic interests in nature, as well as on education through the travellers' involvement in local economic and cultural activities. On the contrary, experiential activities that take place in urban areas have a distinct "theatrical" aspect, as the cities, through their historical background, their architectural reserve, and their often multicultural heritage, become the "stage" on which experiential activities take place. These findings support previous studies on the subject [2,8,21]. Despite the fact that experiential activities do not necessarily imply the involvement of small enterprises only, in our study area, with a dominant mass tourism sector, the experiential product was provided only by small-scale enterprises, which develop such products as their strategic choice, in order to differentiate and, therefore, to address the competitive conventional tourism market. Furthermore, the major public or

corporate stakeholders (e.g., Ministry of Tourism, Greek National Tourism Organization, Greek Tourism Confederation) pointed out that it is still difficult to incorporate experiential products or services in mass tourism and large-scale tourism enterprises.

The qualitative survey highlighted the dominance of small tourism enterprises in the experiential tourism sector. More specifically, most of the interviewees indicated that the shift to experiential activities and, generally, to alternative forms of tourism was the only way for tourism diversification and survival in a competitive tourism market. Innovation is a critical aspect for these enterprises. As described by most interviewees from private enterprises, innovation for them is a unique mix of resources, emphasizing the unknown, and the attractive distinctiveness of the BSB-GR regions, that can be used for the development of an innovative product. This product takes form, consisting of activities that originally belong to various tourism types: rural, gastronomic, cultural, etc. [5]. Such a product is seen to offer unexpected pleasure to tourists, meet their excitement needs, and induce emotions that will tie them to the place-destination and motivate them to come again and/or prolong their stay [21]. The survey indicates that these tourism enterprises need continuous creativity to enhance the tourism experience and to provide tourist destinations with a unique atmosphere [16].

Place composes the physical or imaginary background of experiential tourism. Our findings show that the construction, communication, and management of a place's tangible and intangible qualities is as essential for experiential tourism as it is for the place itself and its branding and marketing [16]. Constructing a positive image or encounter between a place and its users, which is the main target of a place-branding and marketing strategy [14,15], is one of the primary objectives embodied in experiential activities as well. Therefore, this interplay between enhancing the qualities of a place through the development of experiential tourism product and promoting the perceptions and images of a place based on place-marketing and branding strategies is critical. The qualitative survey indicated that the small, locally based enterprises that develop experiential products have the potential to play a major role in promoting and branding destinations. A very interesting aspect is also the fact that the experiential tourism activities that are being developed in non-basic historical landmarks and places in the BSB-GR regions have the potential to channel visitor flow to areas that may not interest visitors independently, thus alleviating pressure on "mass tourism" sites, motivating heritage conservation, and reducing intraregional inequalities by providing socio-economic, cultural, and environmental benefits to disadvantaged localities.

In conclusion, this study has used qualitative findings from the two BSB-GR regions, i.e., the regions of Central Macedonia and Eastern Macedonia and Thrace, to investigate experiential tourism and shed light on the specific features of the experiential tourism product and enterprises. The two BSB-GR regions, both with a well-developed mass tourism profile, offer plenty of material to delve into the development trends and characteristics of experience economy in the field of tourism and its interactions with the tangible and intangible qualities of places. From a theoretical standpoint, the main new knowledge that our study brings is the prominence of the links between three main factors: the experiential product, the small and well-qualified enterprises, and a place's tangible and intangible identities. What our analysis suggests is that these links differentiate "experiential tourism", as based on local culture and heritage, from potential experiential products/services adopted by the conventional mass tourism branches as a result of expected changes in products provided by this form of tourism. Thus, experiential tourism, as a distinct branch of activities, may become part of a vision for local development and local differentiation; in other words, it can offer an opportunity for locales and their branding that is based on local innovation and knowledge.

From the policy and management point of view, our analysis suggests that, although experiential tourism has occupied the relevant debate for more than two decades, it still holds a rather small part of the tourism product and activities, while there is still insufficient knowledge about it on behalf of a great number of stakeholders. Mass tourism seemingly

has, and will continue to have, a great importance for the tourism industry and tourist demand, especially in national economies that are heavily dependent upon tourism, such as the Greek economy. However, many factors, along with unforeseen conditions such as the current COVID-19 pandemic, may intensify transformation processes, with experience becoming an essential part of a journey by integrating multidimensional environmental and cultural parameters to stimulate memorable emotions, education, and creativity.

The existing experiential tourist product, although rather limited, is dynamic and multidimensional. Its further development requires constant feedback as well as adaptation to unforeseen conditions to become sustainable and face abrupt changes. At the same time, it is necessary to strengthen the existing infrastructures and modernize the legislation regarding tourism and its alternative forms under a climate of cooperation and solidarity at all levels [14,16].

The research findings highlight the fact that it is necessary to invest a lot of effort and to mobilize several players to achieve efficient cooperation between essential participants in the sector, and to build an attractive, high-quality experiential tourist product [16]. Alternative operational bodies may be required, such as a Destination Management Organization with deep knowledge of the advantages and special features of each destination. In this way, places and destinations that may not have been considered to have tourism potential, would develop their comparative advantages, and create a high-quality experiential product. To create such a tourist product, it is necessary to build trust among different public and private players that deliver different parts of innovative tourist activities. Further research is also crucial to recognize how the distinct characteristics of places offer potential for physical or imaginary destinations and, hence, contribute to the economic, social, and environmental sustainability of these places. Finally, the need for a strategic plan to promote the potential of experiential tourism at a national, regional, and local level is imperative.

**Author Contributions:** Conceptualization: A.Y., A.A., V.B.-A., A.P. and A.V.; methodology: A.Y., A.A. and V.B.-A.; investigation: A.Y., A.A. and V.B.-A.; writing—original draft preparation: A.Y., A.A., V.B.-A., A.P. and A.V.; writing—review and editing: A.Y., A.A. and V.B.-A.; supervision: A.Y.; project administration: A.P. All authors have read and agreed to the published version of the manuscript.

**Funding:** This research is part of PRO EXTOUR project, which is funded by the European Union in the framework of the ENI CBC Black Sea Basin Programme, with the Grant Contract BSB 1145.

**Informed Consent Statement:** Informed consent was obtained from all subjects involved in the study.

**Data Availability Statement:** Not applicable.

**Acknowledgments:** We are thankful to Dimitris Angelis, graduate student of the Department of Economics, University of Macedonia, for his support during the desktop survey.

**Conflicts of Interest:** The authors declare no conflict of interest.

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
