# Peer review of "Branding Places through Experiential Tourism: A Survey on the Features of the Experiential Product and Enterprises in Greek Regions"

_tourismhosp, doi:10.3390/tourhosp3020028_

Round 1
Reviewer 1 Report
Thank you for submitting your paper to T&H. The paper is timely and does address an interesting question regarding the real development of experiential marketing within the tourism industry and whether it can represent a new attractive feature and a part of. a branding strategy.
The experiential review is interesting, you could add to your review the latest book on. experiential marketing (Frochot Routledge, 2021) that also addresses experiential dimensions, including human relations. I feel that the methodology could be better explained, it is difficult to clearly understand how the authors identified their results. For instance on page 10 how did the authors identified whether providers had " a good understanding of the term" (line 384). What was the definition given to the term by the authors? How did they identify to which extent the providers diverged from it or not?
On page 13, there is an interesting comment about the size of the businesses? Do the authors mean that experiential activities can only be the object of small enterprises?
throughout, it is not always clear how the categories of the tables were identified (figures 3 and 4) particularly, can it be better explained.
The conclusion lacks theoretical implications. The literature review investigated a broad range of academic work, what are the results brings, in terms of new knowledge, for those models?
Also, how was the stakeholders' relevance to the study identified? (page 6 - 235)?
Page 8, some operators are identified as "operate as experiential products" can the authors please be more precise and explain how they identified this. I fill that, from the onset, the authors should describe more precisely how they identified and selected "experiential operators".
Author Response
Dear Reviewer,
Thank you very much for reviewing our paper and providing your constructive comments and valuable suggestions, which gave us the opportunity to improve our paper. Our responses to each of the comments are as follows.
Thank you for submitting your paper to T&H. The paper is timely and does address an interesting question regarding the real development of experiential marketing within the tourism industry and whether it can represent a new attractive feature and a part of a branding strategy.
Thank you very much for your positive feedback and insightful comments.
The experiential review is interesting, you could add to your review the latest book on. experiential marketing (Frochot Routledge, 2021) that also addresses experiential dimensions, including human relations.
We added a reference to this recent and very interesting work both in the introduction and in our conceptual framework.
I feel that the methodology could be better explained, it is difficult to clearly understand how the authors identified their results. For instance on page 10 how did the authors identified whether providers had “ a good understanding of the term” (line 384). What was the definition given to the term by the authors? How did they identify to which extent the providers diverged from it or not?
In the methodology, Section 3.2, p. 7 1st paragraph, we added a brief explanation pointing out that we used open questions to identify stakeholders’ understanding of the term experiential tourism. Also, in the description of results, Section 3 p.11 1st paragraph, we added an explanation how we drew up our conclusion regarding the stakeholder’s perception of this term.
On page 13, there is an interesting comment about the size of the businesses? Do the authors mean that experiential activities can only be the object of small enterprises?
In section 4 Discussion and Conclusions (p.14) we explained further our comment regarding the size of businesses, so as to clarify better our argument.
Throughout, it is not always clear how the categories of the tables were identified (figures 3 and 4) particularly, can it be better explained.
In sections 3.2 (p. 10) and 3.3 (p. 12) we provided respectively an explanation on how the categories of the tables were identified.
The conclusion lacks theoretical implications. The literature review investigated a broad range of academic work, what are the results brings, in terms of new knowledge, for those models?
We worked further our conclusions to highlight the theoretical implications and to indicate the new knowledge our study brings. Accordingly, we made some changes in the abstract so as to highlight and justify better the importance of this study.
Also, how was the stakeholders' relevance to the study identified? (page 6 - 235)?
In the methodology, Section 3.2., p. 6 we clarified better how we evaluated the relevance of the stakeholders. In addition, we clarified the reason that some stakeholders were selected in an ad hoc manner.
Page 8, some operators are identified as "operate as experiential products" can the authors please be more precise and explain how they identified this. I fill that, from the onset, the authors should describe more precisely how they identified and selected "experiential operators".
In fact, the phrase “operate as experiential products” was used wrongly. So, in (now) p. 9 we corrected this phrase and made more clear which ventures we refer to.
Reviewer 2 Report
The paper aims to explores the features of experiential tourism in Greek regions.
The article deals with an interesting topic, where the experiential tourism is highly attractive in recent years. The study is well structured, but some minor revisions could increase the quality of the article.
The aim can be clarified in the abstract. It would be appropriate highlight the gap in the literature and justify the importance of the study.
The introduction is well organised and offers an interesting perspective on the topic of Experiential tourism.
The methodology is well structured but to make it clearer and more intuitive for the reader it would be appropriate to include a figure that summarises the research design.
The results section is well organised and present interesting information.
In the discussion and conclusion section research and management implications could be strengthened
The work is very interesting and well structured, acting on the points mentioned can make the study of higher quality.
Author Response
Dear Reviewer,
Thank you very much for reviewing our paper and providing your constructive comments and valuable suggestions, which gave us the opportunity to improve our paper. Our responses to each of the comments are as follows.
The paper aims to explore the features of experiential tourism in Greek regions.
The article deals with an interesting topic, where the experiential tourism is highly attractive in recent years. The study is well structured, but some minor revisions could increase the quality of the article.
Thank you very much for your positive feedback and insightful comments.
The aim can be clarified in the abstract. It would be appropriate highlight the gap in the literature and justify the importance of the study.
We made some changes in the abstract so as to highlight and justify better the importance of this study.
The introduction is well organized and offers an interesting perspective on the topic of Experiential tourism.
The methodology is well structured but to make it clearer and more intuitive for the reader it would be appropriate to include a figure that summarises the research design.
The results section is well organized and present interesting information.
Thank you very much for your positive feedback and insightful comments. In the methodology we added a figure (Figure 2) to summarize our survey design.
In the discussion and conclusion section research and management implications could be strengthened
We worked further our conclusions to highlight the theoretical implications and to indicate the new knowledge our study brings. We clarified better which are the conclusions from the theoretical and from the policy and management point of view.
The work is very interesting and well structured, acting on the points mentioned can make the study of higher quality.
Thank you very much for your positive feedback and insightful comments. We hope that the changes we made have improved further the quality of our paper.
Reviewer 3 Report
I think the paper is exceptionally well-written and has only minor improvements to make in the formatting of the sentences at the end of the row where there is a cut in wording such as in line 483 experi- then in line 484 ential product. These can be changed using the formatting tools. Please also add references in the methodology section, regarding the grades scale, if available (line 266 to 276)

Author Response
Dear Reviewer,
Thank you very much for reviewing our paper and providing your constructive comments and valuable suggestions, which gave us the opportunity to improve our paper. Our responses to your comments are as follows.
I think the paper is exceptionally well-written and has only minor improvements to make in the formatting of the sentences at the end of the row where there is a cut in wording such as in line 483 experi- then in line 484 ential product. These can be changed using the formatting tools. Please also add references in the methodology section, regarding the grades scale, if available (line 266 to 276)
Thank you very much for your positive feedback and insightful comments.
The cuts in the words made at the end of rows were due to the format of the template. Yet we have corrected them.
We added the reference regarding the grades scale which was missed by mistake as it was from another work of one of the authors of this paper.
We also made the changes suggested in the attached pdf.
Round 2
Reviewer 1 Report
Thank you for resubmitting your paper. I feel that most of the comments have been met and I advise for publication of this manuscript.